# Coumarin Derivatives in Inflammatory Bowel Disease

**DOI:** 10.3390/molecules26020422

**Published:** 2021-01-15

**Authors:** Luiz C. Di Stasi

**Affiliations:** Laboratory of Phytomedicines, Pharmacology, and Biotechnology (PhytoPharmaTech), Department of Biophysics and Pharmacology, Institute of Biosciences, São Paulo State University (UNESP), 18618-689 Botucatu, SP, Brazil; luiz.stasi@unesp.br

**Keywords:** inflammatory bowel disease, coumarin, isocoumarin, Crohn’s disease, ulcerative colitis, glutathione, oxidative stress, complementary therapies, intestinal inflammation

## Abstract

Inflammatory bowel disease (IBD) is a non-communicable disease characterized by a chronic inflammatory process of the gut and categorized into Crohn’s disease and ulcerative colitis, both currently without definitive pharmacological treatment and cure. The unclear etiology of IBD is a limiting factor for the development of new drugs and explains the high frequency of refractory patients to current drugs, which are also related to various adverse effects, mainly after long-term use. Dissatisfaction with current therapies has promoted an increased interest in new pharmacological approaches using natural products. Coumarins comprise a large class of natural phenolic compounds found in fungi, bacteria, and plants. Coumarin and its derivatives have been reported as antioxidant and anti-inflammatory compounds, potentially useful as complementary therapy of the IBD. These compounds produce protective effects in intestinal inflammation through different mechanisms and signaling pathways, mainly modulating immune and inflammatory responses, and protecting against oxidative stress, a central factor for IBD development. In this review, we described the main coumarin derivatives reported as intestinal anti-inflammatory products and its available pharmacodynamic data that support the protective effects of these products in the acute and subchronic phase of intestinal inflammation.

## 1. Introduction

Currently, the search and discovery of new drugs with efficacy, safety, and quality control to prevent and treat non-communicable diseases is a huge challenge for the chemical and pharmaceutical sciences as well as medicine. This task is not only a challenge in the present time but also to guarantee health quality for the next generations. Non-communicable diseases, also known as chronic diseases, are persistent illnesses generally without a pharmacological cure, that tend to be of long duration, requiring long-term and systematic treatment approaches, and the result from multifactorial etiological factors. In general, patients with non-communicable diseases live throughout their lives with several symptoms, continuously using drugs to relieve them. Non-communicable diseases are a group of chronic disorders including cardiovascular diseases, diabetes, multiple sclerosis, obesity, arthritis, asthma, Parkinson’s and Alzheimer’s diseases, cancer, and inflammatory bowel disease (IBD), which have a high impact on the health system. According to a recent study, 56 million people died in 2017 and non-communicable diseases account for more than 73.4% of these global deaths, i.e., 41.1 million people [1]. Based on this, the search for new drug development to relieve symptoms and mainly to prevent non-communicable diseases is an important approach to improve several world health problems and patients’ life quality.

The discovery of new drugs is based on synthetic chemistry, partial synthesis or modification of active molecules of synthetic or natural origin, and bioprospection of natural products, particularly from fungi and plant species. The research with natural products was overshadowed by the advent of the new technologies, synthesis of several chemical active compounds, and international regulatory systems for biodiversity access established by the United Nations Convention on Biological Diversity. However, all current difficulties did not reduce the importance of the world biological biodiversity, particularly from tropical areas, as an inexhaustible and magnificent source of new medicines, which should be carefully and legally studied, respecting international regulatory systems and the traditional knowledge from local communities.

Natural products from plant and fungi origin are the source of several drugs with wide applications and pharmacological importance. Some of these compounds have defined the way of science and modern medicine as well as represented the basis of the treatment of several serious diseases and health problems affecting the world’s people. The antibiotic penicillin discovered in *Penicillium* genus fungi; morphine, an opioid compound useful as a pain reliever, isolated of the opium plant (*Papaver somniferum*), and acetylsalicylic acid, a lead compound of the non-steroidal anti-inflammatory drugs, which is related to salicin, obtained from plants belonging genera *Salix* and *Populus*, are some emblematic examples of the natural products that have changed the history of medicine. Even with advanced modern medicine and biotechnology, the most recent discoveries of lead compounds include two products of plant origin, artemisinin and taxol, an antimalarial and an antineoplastic agent, respectively. The research related to the discovery of artemisinin from *Artemisia annua* received the 2015 Nobel Prize in Physiology or Medicine. Artemisinin completely changes the control of malaria and represents a new class of antimalarial drugs, whereas taxol, isolated from species belonging to the genus *Taxus*, particularly *Taxus brevifolia*, represents a new class of anti-cancer drugs. The number of lead compounds obtained from nature is high, showing that natural products play a key role in human health surveillance and represent the support basis of drug research and discovery.

Plant-based products are rich in several chemical classes of compounds, among which the alkaloids, terpenoids, tannins, and phenol and polyphenol compounds stand out, which are potentially useful to prevent and treat several disorders, particularly non-communicable diseases. Phenol and polyphenolic compounds, one of the most important classes of secondary metabolites from plants, include a plethora of different classes of molecules with high pharmacological value, among which the flavonols, flavanones, flavones, anthocyanidins, xanthones, stilbenes, catechins, quinones, and coumarins may be highlighted. These compounds represent an important source of new molecules with several pharmacological properties and are widespread in vegetables commonly consumed daily as dietary foods and spices. Dietary intake of several plants containing these compounds contributes to the plasma bioavailability of active molecules, which are useful both to improve immune response and act as preventative products for several non-communicable diseases. Nowadays, it has been considered that a properly used nutritional approach might be a part of the treatment of non-communicable diseases, particularly patients with Crohn’s disease and ulcerative colitis, two chronic inflammatory disorders of the gut [2]. The pharmacological properties of phenol and polyphenol compounds against the inflammatory processes of the gut have been exhaustively reported, focusing on flavonoids [3,4], proanthocyanidins and anthocyanins [5,6], and catechins [7]. However, there is a lack of data and analysis of the potential use and application of coumarin and their derivatives as preventative and curative compounds in non-communicable diseases, particularly in inflammatory bowel diseases (IBDs).

In this review, we aim to update and systematize the available knowledge on the pharmacological activities of coumarin derivatives in the various in vivo experimental models of intestinal inflammation and in vitro studies to provide data and insights to further preclinical, clinical, and molecular studies, demonstrating the main and potential active coumarins useful to prevent or treat inflammatory bowel diseases as well as its main mechanisms of action and signaling pathways.

## 2. Coumarin and the Main Coumarin Derivatives

Coumarins, also known as benzopyrones, comprise a class of cinnamic acid-derived phenolic compounds found in fungi, bacteria, and plant species, particularly in edible, medicinal, and spice plants from different botanical families [8]. Coumarins are secondary plant heterocyclic metabolites composed of fused benzene and α-pyrone rings (Figure 1), and they occur widely in different parts of plants, such as roots, seeds, nuts, flowers, and fruits either as heterosides or in free form [8,9]. The term coumarin originated from de name “cumaru” the local name for the Brazilian teak plant (*Dipteryx odorata* Wild.) from the Fabaceae botanical family, in the traditional medicine of the Brazilian Amazon forest. *Dipteryx odorata* is an endemic plant of Central America and the North of South America, widespread in the Amazon Forest region, from which the coumarin was firstly isolated by Vogel in 1820 [10]. Its seeds, named tonka beans, are the natural source of coumarin, a compound widely used by the perfumery companies to replace vanilla, particularly as a fixative and enhancing agent in perfumes as well as added to toilet soap, detergents, toothpaste, tobacco, and alcoholic products [9]. Moreover, the isocoumarin, also recognized as 1*H*-benzopyran, is an isomer of the basic structure of coumarin, in which the orientation of the lactone ring is reversed (Figure 1). From the different substituted groups on the basic structure of isocoumarin, several subclasses and isocoumarin derivatives are also found in plant species such as paepalantine, capillarin, and thumberginol A (Figure 1).

Coumarins are categorized into four main subtype classes of compounds: simple coumarins, furanocoumarins, pyranocoumarins, and the pyrone-substituted coumarins. Simple coumarins are composed of molecules with hydroxyl, alkoxyl, and alkyl substitution patterns on the basic structure and their glucosides [11]. Simple coumarin class represents the main class of coumarin derivatives with intestinal anti-inflammatory properties, particularly esculetin, esculin, 4-hydroxycoumarin, osthole, and 4-methylesculetin (Figure 2).

Furanocoumarins are composed of coumarins derivatives in which a furan ring is fused with the basic structure of coumarin via C6-C7 or C7-C8 [10,11], generating linear furanocoumarins (fusion via C6-C7) such as psolaren, imperatorin, and xanthotoxin or angular furanocoumarins (fusion via C7-C8) such as isobergapten and angelicin (Figure 3). Similarly, in the pyranocoumarins, a six-membered pyran ring is fused with the benzene ring of the basic structure of coumarins via C6-C7 or C7-C8 [10,11]. Decursin (a linear pyranocoumarin) and seselin (an angular pyranocoumarin) are some examples of pyranocoumarins (Figure 3), which have no intestinal anti-inflammatory activity.

Finally, pyrone-substituted coumarins are coumarin derivatives containing different chemical radicals fused with the pyran ring of coumarin [10,11]. Pyrone-substituted coumarins include natural and synthetic coumarins such as warfarin and dicoumarol (Figure 3). Pyrano-substituted coumarins have no intestinal anti-inflammatory activity but comprise some compounds with high pharmacological relevance. Warfarin is a derivative of dicoumarol, a pyrano-substituted coumarin isolated from hay species (*Melilotus alba* and *Melilotus officinalis*) after natural oxidation by several fungi, mainly *Penicillium nigricans* and *Penicillium jensi* found in moldy hay [12]. Dicoumarol and warfarin were first used as rodenticides due to their ability to promote internal hemorrhage in rodents [12]. The anticoagulant properties of dicoumarol and warfarin were the basis for the development of anticoagulant drugs to prevent stroke in patients with cardiovascular diseases, mainly atrial fibrillation and valvular heart disease, and to prevent and treat vein thrombosis and pulmonary embolism [13].

## 3. Inflammatory Bowel Diseases: General Aspects

Inflammatory bowel disease (IBD) consists of Crohn’s disease (CD) and ulcerative colitis (UC), two relapsing chronic inflammatory processes of the gastrointestinal tract, which are part of a group of immune-mediated inflammatory diseases, without a definitive pharmacological treatment and cure [14]. Patients with CD or UC live with several harmful effects in their daily physical, social, and professional activities because these diseases produce limiting effects such as changes in intestinal habits with several evacuations, abdominal pain, diarrhea, bleeding, perianal fistulas and other extraintestinal manifestations.

IBD is a disease that is increasing globally, affecting some 6 to 8 million people in the world and presenting a prevalence rate of 84.3 people (79.2 to 89.9) per 100,000 population in 2017 [15]. Although it is a disorder with low mortality and with a death rate of 0.51 per 100,000 population, IBD is growing exponentially around the world, showing the highest prevalence rates in North America and the United Kingdom and other European countries such as Norway, Poland, and Slovakia, whereas lower prevalence rate has been reported in several countries of Africa, South America, and Southeast Asia [15]. There is a direct relationship between the high prevalence rate of IBD and the industrialization level of a specific country, but the prevalence and incidence rates notably are also rising in newly industrialized countries [16]. The prevalence and incidence rate increment in developing or newly industrialized countries is associated with the industrialization process and migration of population from rural to urban areas, which promote changes in the lifestyle and the people choices related to diet, daily activities, and social behaviors. These changes suggest rates of IBD prevalence and incidence should thrive in parallel to those in the industrialized countries [16].

The IBD etiology is unclear, but several triggering factors have been related to its occurrence and development, including dysregulated immune response, dysfunctional intestinal barrier function, genetic, and environmental aspects. Currently, the pathogenesis of IBD involves a dysregulated autoimmune response and increased intestinal permeability related to gut dysbiosis, which is accelerated by exposure to environmental factors in individuals who have a pre-existing high-risk genotype [17]. The complex and inexact etiology of IBD is a limiting factor in the discovery of new pharmacological and complementary therapies, and the development of preventive strategies useful to define a general protocol of treatment and definitive management of IBD patients. The multifactorial aspects of IBD also explain the high frequency of patients who are refractory to current pharmacological treatments, including conventional drugs such as aminosalicylates, glucocorticoids, immunosuppressants, and biological therapies based on monoclonal antibodies [18]. Current pharmacological treatment of IBD is based on the relief or to create a time of deep remission of symptoms. However, the long-term use of these drugs, produces serious side effects, reducing patient adherence to pharmacological treatment.

IBD also includes a lot of risk factors with an imbricated relationship among them. A series of interactions among risk factors, which do not act in isolation because none of the risk factors alone is sufficient for IBD development, have been suggested [19]. Risk factors for IBD development include intrinsic and extrinsic factors. The intrinsic factors involve genetic predisposition and familiar history as well as the intestinal microbiota, whereas extrinsic risk factors include smoking, appendectomy, hygiene, infections, use of antibiotics and other drugs such as NSAIDs (Non-steroidal anti-inflammatory drugs) and oral contraceptives, a diet with lower fiber and higher fat, vitamin D deficiency as well as lifestyle and social behavior, mainly high stress, sleep privation and lower physical activity [19]. Genetic and epigenetic studies have been extensively used as an important source of data, which are important for a better prediction of IBD course, identification of loci and candidate genes yielding valuable insights into the pathogenesis of IBD and disease pathways, which can be relevant in the clinical practice [20,21].

Intestinal microbiota, which has a key role in the pathogenesis of IBD, is an intrinsic factor that can be modulated by a series of products able to differentially affect distinct microorganisms, including functional food products, mainly probiotic, prebiotic, and symbiotic, natural products such as polyphenol compounds and standardized phytomedicines. In health conditions, intestinal microbiota via fermentation of dietary components produces a series of metabolites, mainly short-chain fatty acids (SCFAs), which are a source of energy for colonocytes and bacteria, and play several protective effects in the body after prompt absorption (Figure 4). On the other hand, the management of extrinsic factors, including changes in lifestyle, social behavior, and diet options as well as a lower exposition to other extrinsic factors can represent an important approach to reduce IBD development.

The combinatory action among genetic predisposition, external environmental factors, and intestinal microbiota is essential to the development of the dysregulated immune response and dysfunctional intestinal barrier [22], which are responsible for IBD development (Figure 4). Both CD and UC patients exhibit a dysfunctional intestinal epithelial barrier with increased permeability as well as an exaggerated immune response in the gastrointestinal tract towards the intestinal microbiota, which is not appropriately controlled, leading to intestinal inflammation [3,22]. The increase of intestinal permeability has been recognized as an early feature of the intestinal inflammatory process, which reduces the intestinal barrier function, a key factor to maintain intestinal homeostasis [23]. In this process, several factors and mediators are involved, particularly the zonulin pathway activation, a key process to control intestinal permeability, suggesting zonulin as a biomarker of gut permeability as well as a key target for the action of intestinal anti-inflammatory products [23,24,25]. The dysregulated immune response includes an innate immune response, the first-line defense against any damage promoted by pathogens, which is mediated by different cell types including macrophages, neutrophils, monocytes, dendritic, epithelial, and endothelial cells [3,22]. These cell types are responsible for phagocytosis, elimination of pathogens, production of several cytokines, and development of barrier and transport functions (Figure 4). The response to pathogens includes prompt participation of the antigen-presenting cells (APCs), which mediate the differentiation of T-cells into effector T helper (Th) cells, including Th1, Th2, and Th17 cell types, and regulator T-cells (Treg) (Figure 4), which are constituents of the adaptative immune response [3,22]. These different cell types are responsible for the synthesis and dysregulated release of a series of immunologic and inflammatory mediators with wide importance in the pathogenesis of IBD, including a plethora of chemokines and cytokines. The pathophysiology of intestinal inflammation is very complex, including a wide number of signaling pathways and endogenous mediators as illustrated in Figure 4, where are particularly indicated the main targets for the action of coumarin derivatives.

Moreover, in the gastrointestinal tract, there is homeostasis of intestinal microbiota with immune cells responsible for a balance of host defense and immune tolerance, which can be shift leading to a dysbiosis process that plays a key role in the pathogenesis of IBD [17]. Although dysregulated immune response associated with both genetic predisposition and intestinal microbiota participates in the intestinal inflammatory process, oxidative stress characterized by an excessive release of reactive oxygen and nitrogen species (ROS/RNS) plays a key role in IBD pathogenesis [26,27]. The excess of oxidative mediators reacts with cell membrane fatty acids, proteins, and DNA impairing their functions (Figure 4). The production of superoxide, the main source of free radicals, is required to kill bacteria, a process that occurs especially in neutrophils and other cell types such as epithelial cells from the intestine. From superoxide production, several free radicals are produced via nitric oxide synthase and glutathione-related enzymes (Figure 4).

## 4. Intestinal Anti-Inflammatory Activity of Coumarin Derivatives

The dysregulated immune and oxidative response triggered by intestinal inflammation is an imbricated and complex interaction involving a series of endogenous mediators from different signaling pathways and receptors, such as nuclear factor-kappa b (NF-κB), nuclear factor erythroid 2 (NEF2)-related factor 2 (Nrf2), peroxisome proliferator-activated receptor gamma (PPAR-γ), pregnane X receptor (PXR), hypoxia-inducible factor (HIF), several enzymes, especially cyclooxygenase 2 (COX-2), mitogen-activated protein kinases (MAPKs), and HIF-prolyl hydroxylases (PHDs) as well as mediators of intestinal epithelial barrier function such as zona occludens 1 (ZO-1), occludin, mucins (MUC1, MUC2, MUC 3, MUC4), and E-cadherin. These different pathways will be discussed in this review to explain the effects of some coumarin derivatives and their partially elucidated mechanisms of action. However, until now the majority of coumarin derivatives produce intestinal anti-inflammatory activity acting as modulators of oxidative stress and immune response, whereas a little number of coumarin derivatives were reported to act by other signaling pathways, which are also partially involved in oxidative stress modulation.

The intestinal anti-inflammatory activity of different coumarin derivatives was described using different experimental models of intestinal inflammation, mainly trinitrobenzene sulphonic acid (TNBS) and dextran sodium sulfate (DSS) in rats or mice as well as several in vitro studies with distinct cell types. Although coumarin is a class of natural and synthetic compounds with high chemical diversity, the intestinal anti-inflammatory has been limited to three subclasses of coumarin derivative, i.e., isocoumarins having one active compound namely paepalantine, a lot of simple coumarin derivatives, and the furanocoumarin imperatorin.

### 4.1. Effects of Coumarin Derivatives on Oxidative Stress

A recent review reported that different antioxidant compounds act by six general mechanisms: inhibiting free radical production by activated oxygen metabolites, changing the structural organization of free radical, producing a local decrease of oxygen concentration, interacting with organic radicals, chelating metal ions, and converting peroxides to stable and inactive products [28]. These processes reduce the availability of free radical species, improving the response against oxidative stress, which are the main properties of the coumarin derivatives with intestinal anti-inflammatory activity.

Oxidative stress is considered an imbalance in which excessive levels of oxygen free radicals such as reactive oxygen species (ROS) and reactive nitrogen species (RNS) are present in the biological system in the face of inadequate availability of the antioxidants, which are capable to destroy these harmful products from metabolic processes [29]. A free radical is a molecule of the normal metabolism of oxygen with an unpaired or odd number of electrons, which is highly reactive and able to react with lipids from the cell membranes, proteins, and nucleic acids, affecting their structure and functions. In an inflammation process, ROS and RNS production is a prompt defense response of the body to kill several invading pathogens as well as to regulate the immune response via pro-inflammatory chemotaxis induction into the site of inflammatory processes as well as modulate the interactions and activation of the immune cell types [30]. However, when occurs an imbalance between free radical production and endogenous antioxidant response, the reaction of reactive oxygen and nitrogen species with host lipids, proteins, and nucleic acids generate oxidative stress, triggering a series of molecular and cellular events such as tissue damage and fibrosis [30], which are related to the origin and development of several chronic diseases, including IBD.

The reactive oxygen species and the endogenous mediators of antioxidative response are represented in Figure 5. The main reactive species in the biological system responsible for oxidative stress is the superoxide radical anion (O_2_^•−^), which is formed by the addition of a single electron in molecular oxygen (O_2_) by the action of reduced nicotinamide adenine dinucleotide phosphate (NADPH) oxidase (NOX) enzyme [26,27]. O_2_^•−^ is the source of hydrogen peroxide (H_2_O_2_), formed by the action of superoxide dismutase (SOD) enzyme, which catalyzes the dismutation of O_2_^•−^ into H_2_O_2_. Although H_2_O_2_ is not a free radical, it is a molecule highly reactive [29,30] and represents a key substrate for free radical production. H_2_O_2_ via Fenton reaction generates hydroxyl radical (OH^−^), which is the most reactive and harmful metabolite of oxygen metabolism. Simultaneously, H_2_O_2_ by the action of myeloperoxidase (MPO) produces hypochlorous acid (HOCl), a potent antioxidant with antimicrobial activity and useful as a defense against several infectious pathogens [30,31].

MPO is the most toxic enzyme found in the granules of neutrophils and monocytes, two important cell types that participate in the intestinal inflammatory process with a key role in the innate immune response to pathogens [30]. MPO generates reactive intermediates, inducing oxidative peroxidation of lipids, and proteins and DNA damage. During the inflammatory response, MPO is released from neutrophils and monocytes, catalyzing the formation of HOCl from H_2_O_2_ (Figure 5). HOCl produced by the action of MPO influences the conversion of glutathione (GSH) to oxidized glutathione (GSSG), which disrupts the cellular redox balance, reducing the antioxidant GSH pool (Figure 5) and increasing the susceptibility to oxidative stress [30]. Generally, MPO activity is strongly increased in experimental models of intestinal inflammation, whereas simultaneously is possible to observe a depletion of GSH [32].

GSH, a tripeptide formed by glutamic acid, cysteine, and glycine, is the key endogenous antioxidant that participates in antioxidant response. GSH is produced by a reaction with two steps (Figure 5). Firstly, a residue of glutamic acid (Glu) binding with cysteine via γ-glutamylcysteine ligase (γ-GCL) action, producing γ-glutamylcysteine [33]. Secondly, γ-glutamylcysteine reacts with a residue of glycine under the action of glutathione synthetase (GS) enzyme to produce GSH [33]. GSH participates in antioxidant response acting as a free radical scavenger, reducing dehydroascorbate to ascorbate, which regenerates α-tocopherol from the α-tocopherol radical oxidation, and serving as a co-substrate for several antioxidant enzymes, mainly glutathione S-transferase (GST) and glutathione peroxidase (GPx) [34]. High pools of GSH is essential to modulate oxidative stress, and GSH also can be regenerated by oxidation of its disulfide-oxidized dimer (GSSG) by the action of glutathione reductase (GR) (Figure 5). This reaction occurs at expense of the reduction of NADPH from the pentose phosphate pathway [34].

The modulation of oxidative stress for the body’s defense against free radical tissue damage and macromolecules oxidation is modulated by antioxidants, which are classified into nonenzymatic antioxidants consisting of micronutrient components, and enzymatic endogenous system [17,29]. The nonenzymatic antioxidant system includes several small molecules, mainly GSH, vitamin E, vitamin C, β-carotene, retinol, uric acid, and ubiquinol as well as several microelements like selenium, iron, zinc, copper, and manganese [29]. Vitamins act as donors and acceptors of ROS and the micronutrients act as cofactors, which regulate the activities of the antioxidant enzymes [29]. Endogenous enzymatic antioxidants involve superoxide dismutase (SOD), catalase (CAT), glutathione peroxidase (GPX), glutathione S-transferase (GST), glutathione reductase (GR), and γ-glutamyl transferase (γ-GT) enzymes [32], which act by different pathways to reduce the free radical availability and control oxidative stress.

In this process, CAT can break down two H_2_O_2_ molecules generating one molecular oxygen and two molecules of water [35], reducing H_2_O_2_ pool availability, which is used as a substrate for the production of OH^−^ (Figure 5). In association with SOD and CAT, GPX, a dependent enzyme of micronutrient selenium, plays an important role in the reductions of H_2_O_2_ and lipid peroxides (LOOH) to produce water and lipid alcohol (LOH), contributing to modulation of oxidative stress and avoiding direct tissue damage [36]. Moreover, GST can catalyze the transfer of a GSH group to organic and inorganic electrophiles, reducing these compounds into unreactive products [34].

In another reaction pathway (Figure 6), nitric oxide synthase (NOS) controls the reaction of O_2_^•−^ with nitric oxide radical (NO^•^) to produce free radical peroxynitrite (ONOO^−^), which is responsible for nitrosylation of proteins and oxidation of lipoproteins [31]. NO^•^ is produced via enzymatic oxidation of L-arginine to L-citrulline by the action of constitutive and inducible NOS. Besides the mediator in blood pressure, NO^•^ participates in the immune and inflammatory responses with biocidal activity against several microorganisms and induces damages on the proteins and DNA [31,37]. Toxic and oxidative effects of NO^•^ results from its oxidation, generating highly reactive species, such as nitrite (NO_2_^−^) and peroxynitrite (ONOO^−^).

NO_2_^−^ is produced by NO^•^ autooxidation forming nitrous anhydride (N_2_O_3_), an intermediate in this conversion recognized as a potent nitrosating agent [31], which can also be used to produce nitrogen dioxide by the action of the MPO enzyme (Figure 6). Carbon dioxide (CO_2_) reacts catalytically with ONOO^−^ to produce nitroperoxycarbonate (ONOOCO_2_), which via homolysis of the O-O bonds, carbon trioxide (CO_3_^•−^), and nitrite dioxide (NO_2_^•^) radicals are produced [31]. Moreover, when ONOO^-^ decomposes in the absence of CO_2_, the NO_2_^•^ and OH^•^ radicals production take place, whereas, in the presence of CO_2_, CO_3_^•−^ and NO_2_^•^ radicals are produced, and in this process (Figure 6). MPO also participate, affecting tyrosine nitration when NO_2_^−^ is used as a co-substrate, with consequent production of NO_2_^•^, a reactive free radical [31]. Based on its diverse action as a marker of neutrophil infiltration, inflammatory process, and oxidative stress, MPO represents a potential target for the development of synthetic and natural compounds against several diseases, including atherosclerosis, acute coronary syndromes, ischemic heart disease, and IBD [30,38,39].

The major regulator of the endogenous antioxidant system is the nuclear factor erythroid 2 (NEF2)-related factor 2 (Nrf2) that protects cells from several stressors agents, such as ROS, RNS, and environmental damage [40]. In physiological conditions, Nrf2 binds with cullin 3 (cul3) and Kelch-like ECH-associated protein 1 (keap1), a key repressor of the Nrf2 signaling pathway (Figure 7), preventing the translocation of Nrf2 to the nucleus [41]. This complex, after ubiquitination, promotes Nrf2 degradation via proteolysis. Under oxidative stress conditions, Nrf2-keap1 complex is uncoupled and a free Nrf2 is translocated into the nucleus, where binds with small Maf (sMaf) proteins [42]. The heterodimer binds with antioxidant response elements (ARE) target genes (Figure 7), regulating the expression of several antioxidant-related endogenous genes, including the enzymes CAT, GPX, SOD, GST, γ-GCL, GR, NADPH quinone oxidoreductase, and heme oxygenase [41]. Based on this, Nrf2 is a key mediator of the antioxidant defense system as well as an important target for the action of new synthetic and natural compounds, including coumarin derivatives such as esculetin.

#### 4.1.1. The Isocoumarin Paepalantine

Paepalantine (9,10-dihydroxy-5,7-dimethoxy-1*H*-naptho(2,3c)pyran-1-one), was the first plant-derived coumarin studied in an experimental model of intestinal inflammation [43]. Paepalantine (Figure 1) is an isocoumarin previously isolated from the capitula of the Brazilian endemic *Paepalanthus bromelioides* plant from the Eriocalulaceae botanical family [44], which produced protective effects in the acute and relapse phases of the intestinal inflammation induced by TNBS in rats [43]. The protective effects observed after oral administration of the 5 and 10 mg/kg were similar to those promoted by the 25 mg/kg of sulphasalazine, a 5-aminosalicylate currently used to treat human IBD, i.e., paepalantine produced intestinal anti-inflammatory activity at doses 2.5 and 5.0-times lower than a reference drug [43]. Intestinal anti-inflammatory activity of the paepalantine was related to prevention of the GSH depletion (Figure 5) and inhibition of the colonic NOS activity (Figure 6), which was upregulated by the inflammatory process, suggesting that intestinal anti-inflammatory activity is related to its antioxidant properties [43]. Paepalantine also inhibited HOCl production in rat neutrophils, reducing oxidative stress by the inhibition of MPO activity and scavenging HOCl (Figure 5) [45]. Moreover, the antioxidant properties of paepalantine were evidenced by its potent scavenging properties in the 1,1-diphenyl-2-picrylhydrazyl (DPPH) and superoxide radicals as well as by its ability to protect mitochondria from hydroperoxide accumulation and mitochondrial membrane lipid peroxidation [46,47]. The inhibition of iNOS activity by paepalantine was recently corroborated through in vitro studies with LPS-stimulated macrophages [48]. In this study, paepalantine binds with the NOS enzyme through several structural amino acids just on the active site of the enzyme, reducing its enzymatical activity [48].

#### 4.1.2. Coumarin and 4-hydroxycoumarin

Intestinal anti-inflammatory activity of coumarin and its derivative 4-hydroxycoumarin (Figure 2) were evaluated in the acute and subchronic phases of the TNBS-induced intestinal inflammation model in rats [49]. In this study, damage score and extension of tissue lesion induced by TNBS were significantly reduced after oral administration of coumarin (25 mg/kg) and 4-hydroxycoumarin (10 and 25 mg/kg), and these protective effects were accompanied by a counteraction of GSH depletion and inhibitory MPO activity (Figure 5) [49]. Although 4-hydroxycoumarin produced effects at lower doses when compared with coumarin, is not possible to suggest that the OH substitution in C4 was directly related to the improvement of its effects in the acute or sub-chronicle protocols of intestinal inflammation induced by TNBS in rats.

#### 4.1.3. Esculetin (6,7-dihydroxycoumarin) and 4-methyl Esculetin

Esculetin and 4-methylesculetin (Figure 2) oral administration produced antioxidant protective effects in rats with TNBS-induced intestinal inflammation [50]. While esculetin counteracted GSH depletion at the dose of 10 mg/kg, with no effects on MPO activity, 4-methylesculetin produced significantly positive effects on the GSH levels (2.5 and 5 mg/kg) and MPO activity (5 and 10 mg/kg) in the acute phase of intestinal inflammation (Figure 5). In the sub-chronicle protocol, when coumarins were administered after induction of intestinal inflammation, the effects of 4-methylesculetin were evidenced on the GSH level and MPO activity, whereas esculetin was inactive to restore the basal value of these mediators [50]. Moreover, the inhibitory concentration of 4-methylesculetin on the lipid peroxidation membranes was approximately twice lower than esculetin [50]. A comparative analysis of data suggesting that 6,7-dihydroxylated coumarins when substituted at C4 with a methyl group had an improvement of effects on the MPO activity. Reduction of damage score and MPO activity was also demonstrated after the intrarectal administration of esculetin at 100 and 200 µM in rats with intestinal inflammation previously induced by TNBS [51]. The antioxidant property of esculetin was also corroborated by its action inhibiting iNOs activity (Figure 6) and modulating Nrf2 (Figure 7) signaling pathway [51,52].

Besides intestinal anti-inflammatory activity related to antioxidant properties counteracting GSH depletion, inhibiting MPO activity and lipid peroxidation [50], the effects of 4-methylesculetin (6,7-dihydroxy-4-methylcoumarin) in acute and subchronic phases of TNBS-induced intestinal inflammation was evaluated in comparison with effects of sulphasalazine and prednisolone in rats as well as in RAW264.7, Caco-2 and splenocytes culture cells [53]. Similar to previously reported, 4-methylesculetin improved clinical, histopathological, and biochemical parameters, such as GSH levels and MPO activity (Figure 5) in both acute and subchronic phases of the TNBS-induced intestinal inflammation model [53]. In a recent study in the DSS-induced intestinal inflammation in mice, 4-methylesculetin also improved histopathological indicators of intestinal inflammation, reduced MPO activity, and markedly counteracted GSH depletion [54].

Considering that the intestinal anti-inflammatory activity of 4-methylesculetin was closely related to several mediators of oxidative stress, an interesting study was carried out to investigate the molecular mechanisms involved in these antioxidant properties [32]. In the TNBS model of intestinal inflammation, treatments with 5 and 10 mg/kg methylesculetin significantly decreased damage score, lesion extension, and diarrhea incidence [32]. These protective effects were accompanied by an inhibition of the MPO and GPx activities and simultaneous increment of GST and GR activities, with no effects on the SOD and CAT activities [32]. Moreover, treatment with 4-methylesculetin was able to prevent downregulation of GR and Nrf2, with no effects on the GRX, GST gene expression, suggesting that GR is a target enzyme for the action of 4-methylesculetin. Molecular interaction between 4-methylesculetin and GR using UV-vis absorbance spectroscopy, fluorescence measurements, saturation transfer difference nuclear magnetic resonance, and computational modeling were performed to identify this interaction [32]. These analyses showed that 4-methylesculetin forms a complex with GR with more than one binding site close to the FAD cofactor, which was reduced by NADPH, whereas equivalents were transferred to a redox-active GSSG, stabilizing the 4-methylesculetin-GR complex with a consequent increment of the GR activity [32]. Based on this, authors demonstrate that 4-methylesculetin acts by different antioxidant mechanisms, i.e., controlling the imbalance between MPO activity and GSH production with an increment of GSH availability, upregulating the GST activity with consequent increase of electrophiles inactivation, upregulating GR activity via stabilization of its enzymatic activity, and upregulating Nrf2 expression that leads to a GR regeneration with consequent GSH maintenance levels [32].

#### 4.1.4. Daphnetin (7,8-dihydroxycoumarin)

A comparative study with several coumarin derivatives in TNBS-model of intestinal inflammation daphnetin (Figure 2) demonstrated a protective effect of daphnetin (Figure 2) in intestinal inflammation after oral administration of the lower doses (2.5 and 5.0 mg/kg) [55]. Daphentin counteracted GSH depletion and inhibited MPO activity as well as showing a potent ROS scavenging property (Figure 5) [55]. Among coumarin derivatives, daphnetin is one of the most studied compounds, with a series of pharmacological activities that corroborate its use in the inflammatory process, mainly acting on the oxidative stress and other signaling pathways of the intestinal inflammatory process, which it will bellow discussed. Antioxidant and anti-inflammatory activities have been reported by different studies, in which daphnetin was reported as a potent antioxidant compound inhibiting lipid peroxidation, scavenging free radical generation, and upregulating the Nrf2 signaling pathway [8,56,57].

#### 4.1.5. Esculin (7-hydroxy-6-O-glucosylcoumarin)

Esculin (Figure 2) promoted protective effects on the DSS- and TNBS-induced intestinal inflammation, counteracting GSH depletion, and inhibiting MPO activity [55,58]. Esculin relieved intestinal inflammatory clinical indicators and histopathological damage promoted by DSS, effects that were accompanied by a downregulation of iNOS expression [58]. In vitro studies with RAW264.7 cells stimulated by LPS demonstrated esculin reducing NO generation as well as the gene expression and protein level of iNOS [58]. Several studies corroborated the antioxidant properties of esculin and its use in different inflammatory processes, mainly acting as a potent scavenging agent [8,56], reducing MPO activity [56], NO production, and iNOS levels [59], as well as markedly activated the Nrf2 signaling pathway related to oxidative stress [60,61].

#### 4.1.6. Other Simple Antioxidant Coumarin Derivatives

A comparative and preliminary study using several simple coumarins with different substitutions in the basic ring of coumarins, including scopoletin, scoparone, fraxetin, 4-methylumbelliferone, esculin, and daphnetin (Figure 2) demonstrated differential intestinal anti-inflammatory and antioxidant properties in a TNBS-induced intestinal inflammation in rats [55]. Among these coumarin derivatives, 4-methylumbelliferone produced no effects on the clinical (damage score, extension of lesion, diarrhea, and length/weight colon ratio) and biochemical parameters such as GSH level and MPO activity. Oral administration of scopoletin (5 and 25 mg/kg), scoparone (5 and 10 mg/kg), and fraxetin (5 and 10 mg/kg) were able to counteract GSH depletion induced by intestinal inflammation with no effects on the MPO activity [55]. On the other hand, oral administration of 25 mg/kg of esculin and 2.5 and 5.0 mg/kg of daphnetin counteracted GSH depletion and inhibited MPO activity, showing daphnetin with a protective effect against intestinal inflammatory process at lower doses when compared with the other coumarin derivatives [55]. Although all coumarin derivatives acted as a radical scavenger, only fraxetin and daphnetin inhibited in vitro assay of lipid peroxidation in the cell membrane with lower inhibitory concentrations [55]. The results corroborate the hypothesis that dihydroxylated coumarins with vicinal diol functionality such as fraxetin, esculetin, and daphnetin exhibit potent ROS scavenging when compared to other coumarin derivatives [8,55].

Osthole (Figure 2) in dinitrobenzene sulphonic acid model of intestinal inflammation improved the histopathological damage and some clinical indicators of intestinal inflammation and acted as an antioxidant product, reducing malondialdehyde levels and MPO activity, increasing GPX, CAT, SOD, and GST levels, and counteracting GSH depletion [62]. In the DSS-model of intestinal inflammation in mice osthole showed protective effects on intestinal inflammation improving clinical parameters and histological damages as well as reducing MPO activity and downregulating colon TNF-α and serum TNF-α levels [63].

Antioxidant simple coumarin derivatives with intestinal anti-inflammatory activity also include isomeranzin, auraptene, and collinin (Figure 2). Isomeranzin was reported as able to inhibit NO release in RAW264.7 cells [64], whereas auraptene and collinin intestinal anti-inflammatory effects were associated with reduced iNOS levels [65]. The intestinal anti-inflammatory activities of isomeranzin, auraptene, and collinin were related to other signaling pathways of the inflammatory process [64,66].

### 4.2. Effects of Coumarin Derivatives on Aarachidonic Acid Metabolism

Several metabolites of the arachidonic acid metabolism pathway have pro-inflammatory properties, indicating that inhibitory action on these metabolites production can be an important target for the development of intestinal anti-inflammatory drugs. Arachidonic acid is produced from membrane phospholipids by the action of several phospholipases, manly phospholipase A2 (Figure 8). The metabolism of arachidonic acid includes several enzymes, mainly cyclooxygenase 1 and 2 (COX-1 and COX-2), and lipoxygenase 5 and 12 (LOX-5 and LOX-12), which play a relevant role in intestinal inflammation [67]. COX-1 is constitutively expressed in several cell types and produces diverse eicosanoids such as thromboxane A2 (TXA_2_) and prostaglandins I2 (PGI_2_), which have platelet and cytoprotective effects, respectively. COX-2 is a cyclooxygenase induced under inflammatory stimuli and the main source of pro-inflammatory prostaglandins, such as PGI_2_ and PGE_2_, and its inhibition by different chemical agents has been considered beneficial to control the intestinal inflammatory process [3,67]. On the other hand, lipoxygenases, mainly LOX-5 is a key enzyme for the production of leukotriene B4 (LTB_4_), the major pro-inflammatory metabolite of arachidonic acid that contributes to the perpetuation of intestinal inflammation [68].

Some coumarin derivatives produced different inhibitory action on arachidonic acid metabolism, improving response against intestinal inflammation. Esculetin was able to reduce the COX-2 levels in the colon of rats with intestinal inflammation induced by TNBS [51] as well as inhibited LTB_4_ and TXB_2_ generation via an inhibitory action on the LOX-5 activity [8,69]. Similar effects were reported to esculin, daphnetin, osthole, imperatorin, auraptene, collinin, and fraxetin [8,63,65,69,70,71,72].

### 4.3. Effects of Coumarin Derivatives on the Immune Response

The modulation of the immune response has been reported as an important action of coumarin derivatives to control intestinal inflammation in experimental models and in vitro studies. Generally, the available data indicated the ability of several coumarins produce effects on the production and release of immune mediators, but the general mechanism of action to produce these responses was not fully investigated. However, several intestinal anti-inflammatory coumarin derivatives described probably modulated the immune response acting on the other signaling pathways here described, particularly those including nuclear signaling pathways.

Besides the antioxidant properties, paepalantine (Figure 1) was demonstrated to inhibit the production of pro-inflammatory cytokines TNF-α and IL-6 in human gastric carcinoma cells and murine macrophages RAW264.7 line [48]. Although in the acute phase of the TNBS model, 4-methylesculetin (Figure 2) produced no effects on the IL-1β, TNF-α, MMP-2, and MMP-9 protein levels, in vitro studies demonstrated that 4-methylesculetin inhibits the production of IL-1β in LPS-stimulated RAW264.7 cells, IL-8 in IL1-β-stimulated Caco-2 cells, and INF-γ and IL-2 in concanavalin a-stimulated splenocytes [53]. 4-methylesculetin treatment in DSS-model of intestinal inflammation reduced IL-6 colon levels, with no effects on the IL-17 and TNF-α colon levels [54]. Esculin (Figure 2) relieved intestinal inflammatory clinical indicators and histopathological damage promoted by DDS, effects that were accompanied by a downregulation of IL-1β, TNF-α, and reduction of IL-1β and TNF-α protein levels [59]. Esculetin (Figure 2) in a model of psoriasis-like skin diseasedramatically suppressed pro-inflammatory cytokine releases such as TNF-α, IL-6, IL-22, IL-23, IL-17α, and INF-γ [73]. A recent and interesting study in mice treated with daphnetin was carried out using different approaches to describe the protection of daphnetin (Figure 2) on the DSS-induced intestinal inflammation model [74]. The evaluation of immune and inflammatory response in this experimental model demonstrated daphnetin avoid intestinal inflammation progression, which was related to an improvement of histopathological damage induced by DSS and modulation of pro-inflammatory mediators, downregulating colon TNF-α, IL-6, IL-1β, IL-21, IL-23, CXCL1, and CXCL2 expression, and increasing IL-10 [74]. Osthole at 100 mg/kg by intraperitoneal route attenuated several clinical indicators of the intestinal inflammation as well as the histopathological lesions and alterations induced by TNBS [75]. The protective clinical effects were accompanied by a significant reduction of IL-1β, TNF-α, IL-6, CXCL10, and COX-2 gene expression as well as by an improvement of the intestinal barrier function, upregulating claudin-1 and ZO-1 mRNA [75]. In another set of evaluations using a model experimental of intestinal inflammation induced by dinitrobenzene sulphonic acid, osthole reduced TNF-α and increased IL-10, with no effects on the INF-γ levels [62]. Isomeranzin treatment with an oral dose of 30 mg/kg, isomeranzin attenuated several clinical and histopathological indicators of DSS- and TNBS intestinal inflammation as well as decreased serum IL-6 and TNF-α expression and colon IL1-β, IL-6, TNF-α, and iNOS mRNA expression [64].

### 4.4. Effects of Coumarin Derivatives on the Nuclear Signaling Pathways

Several drugs and natural products, including coumarin derivatives, produce intestinal anti-inflammatory activity acting on the transcription factors, nuclear receptors, and enzymes related to the inflammatory response, particularly nuclear factor-kappa b (NF-κB), peroxisome proliferator-activated receptor gamma (PPAR-γ), mitogen-activated protein kinases (MAPKs), pregnane X receptors (PXRs), rexinoid X receptors (RXRs). Other receptors such as glucocorticoid receptor (GR), farnesoid X receptor (FXR), estrogen receptor (ER), liver X receptor (LXR) regulate the inflammatory response in several diseases such as atherosclerosis, obesity, diabetes, multiple sclerosis, cancer, and IBD [76], showing that these nuclear signaling pathways are key targets for the action of new intestinal anti-inflammatory compounds.

#### 4.4.1. NF-κB and PPAR-γ Signaling Pathways

The transcription factor kappa B (NK-κB) has a central role in the intestinal inflammatory processes, triggering a high pro-inflammatory cytokines production. NK-κB signaling pathway (Figure 9) can be activated either canonical or noncanonical pathways, however, the majority of products and studies were focused on the canonical signaling pathway [77,78]. In the canonical NK-κB signaling pathway, the NK-κB heterodimer consists of the subunits p50 and p65/Rel A, which is inactive in the cytoplasm when binding with inhibitors of protein kappa B (IκB). The IκB inhibitory enzymatic complex (IKK) is composed of a regulatory IKK gamma (IKKγ) subunit and two enzymatically active subunits, IKK alpha (IKKα) and beta (IKKβ) [79]. In the canonical NK-κB signaling pathway, IKK activation occurs by specific membrane ligands such as cytokines, bacteria, bacteria metabolites, viruses, and growth factors [80]. Under this stimulation, IKKβ is activated leading to IκB phosphorylation with consequent ubiquitination and proteasome degradation [74,75,76,77], whereas IKKα is phosphorylated to activate noncanonical NK-κB pathway (Figure 9), causing p100 processing and formation of p52/RelB dimers instead of p50 and p65/Real [77,80]. The released NK-κB is promptly translocated into the nucleus to activate specific response elements in DNA, triggering a transcriptional activity with high production of diverse inflammatory mediators, mainly TNF-α, IL-1β, COX-2, IL-6, IL-8, IL-12, and IL-23 (Figure 9) [77,78,79,80,81].

In several studies, the peroxisome proliferator-activated receptor gamma (PPAR-γ) has been associated with the inflammatory response coordinated by the NF-κB signaling pathway [82,83]. PPAR-γ and other PPARs, such as PPARα and PPARβ are a group of nuclear receptors that modulates glucose metabolism, adipogenesis, fatty acid synthesis as well as inhibit the NF-κB inflammatory response [83]. It has been reported that PPAR-γ deletion induces an increment of the inflammatory process in the DSS model of intestinal inflammation, whereas its activation represses the nuclear localization of NF-κB [81,83], showing NF-κB-dependent response of PPAR-γ as a target for the action of intestinal anti-inflammatory compounds. PPAR-γ is a heterodimer complex with retinoid X receptor alpha (RXRα) generally binding with a co-repressor and expressed in several cells that participates in the intestinal inflammatory response such as dendritic cells, macrophages, and monocytes [83]. Under receptor activation by ligands, the co-repressor molecule displaced, whereas PPAR-γ/RXRα free complex binding with coactivator molecules [83].This activated complex binding to PPAR-γ response elements (PPRE) inducing transcription and protein synthesis (Figure 9). It has been reported that activated PPAR-γ/RXRα/coactivators complex can also bind with NF-κB repressing its transcriptional function with consequent reduction of pro-inflammatory cytokines production and consequent anti-inflammatory effects [81,82,83,84,85]. Several exogenous and endogenous PPAR-γ ligands have been reported, including fatty acids (linoleic, palmitoleic, and oleic acids), eicosanoids (eicosapentaenoic and docosahexaenoic acids, and prostaglandins), thiazolidinediones (rosiglitazone and pioglitazone), non-steroidal anti-inflammatory drugs (indomethacin and ibuprofen) as well as short-chain fatty acids, mainly butyrate and propionate, which are produced from the fermentative process of dietary fiber and other food products by intestinal microbiota [83,86,87].

Several coumarin derivatives produced intestinal anti-inflammatory activity acting on the NF-κB and PPAR-γ signaling pathway in both in vivo and in vitro studies. Antioxidant esculetin (Figure 2) treatment of the human pancreatic cell lines resulted in a significant reduction of NF-κB levels via its binding with Keap1 regulator of the Nrf2 signaling pathway, attenuating the NF-κB activation [88]. Moreover, esculetin reduced the NF-κB p65 levels in the cell nucleus of human NB4 leukemic cell lines [52].

Esculin (Figure 2) was also able to decrease nuclear protein levels p65 from NF-κB signaling pathway both rectal tissue from the animal with DSS-induced intestinal inflammation and RAW264.7 cells [59]. Moreover, esculin suppressed the phosphorylation of IκBα, the major step of NFκB accumulation in the cell nucleus [58]. Finally, the authors elegantly demonstrated that inhibition of NFκB activation by esculin was partially mediated by the PPAR-γ stimulation, promoting nuclear localization of PPAR-γ (Figure 9) and the regulation on NFκB activation [58]. Osthole (Figure 2) was also evaluated in the DSS-model of intestinal inflammation in mice and its protective effects on intestinal inflammation were related to a downregulation of the NFκB p65 and IκB gene expression with a simultaneous effect increasing IκBα protein levels (Figure 9), suggesting that osthole at doses of 20 mg/kg inhibited NFκB activation [63]. Isomeranzin (Figure 2) treatment reduced the phosphorylation of ERK and p65 in DSS- and TNBS-induced intestinal inflammation models. In vitro studies was demonstrated isomeranzin inhibiting NF-κB activation via prevention of TRAF6 ubiquitination, a signal transductor of NF-Κb [64].

#### 4.4.2. MAPK Signaling Pathway

Mitogen-activated protein kinase (MAPK) signaling exerts several effects on cell function, including cell growth, proliferation, differentiation and survival, as well as is closely implicated in IBD, influencing the progression and perpetuation of intestinal inflammation [89,90]. MAPK activation is a response to several extracellular stimuli such as environmental stress, hormones, growth factors, and cytokines that via different kinase receptors, pathogen-associated molecular patterns, and danger-associated molecular patterns recruit pattern recognition receptors to induce a cell response [89,91]. The MAPK signaling pathway includes three groups of protein kinases, i.e., the extracellular signal-regulated kinases (ERKs), the c-Jun N-terminal kinases (JNKs), and the p38 MAPKs [89]. Phosphorylation, which occurs in a specific amino acid sequence of each group of MAPK, is pivotal for their activation [90]. Each group of MAPK is activated by different kinase pathways using distinct interlinked kinase components, as elegantly described [86]. After activation, MAPK is translocated to the nucleus to phosphorylate a series of transcription factors responsible for the expression of several genes and protein synthesis of mediators related to the inflammatory response [89,90,91].

MAPK signaling pathway has been related to the action of intestinal anti-inflammatory drugs, including aminosalicylates, glucocorticoids, and immunomodulators [92] as well as the target for the action of several coumarin derivatives. In mouse peritoneal macrophages, osthole (Figure 2) treatment significantly attenuated the production of the pro-inflammatory cytokines via suppressive effects on the p38 phosphorylation, suggesting its protective effects in TNBS-induced intestinal inflammation was related to the MAPK signaling pathway [75]. A similar evaluation of osthole was performed using the dinitrobenzene sulphonic acid model in rats, DSS-induced intestinal inflammation in mice, and murine macrophages [62,63]. Oral administration of 50 mg/kg of osthole reduced phosphorylation of the MAPK/p38 protein, promoting protective effects in the intestinal inflammatory process [62,63]. In vitro studies demonstrated osthole significantly reduced phosphorylation of p38/MAPK with no effects on the phosphorylation of the ERK and JNK [60], corroborating the data previously reported [62]. Differentially, isomeranzin (Figure 2) treatment in LPS-stimulated murine macrophages reduced phosphorylation of ERK with no effects of the JNK and p38 MAPKs [64].

#### 4.4.3. HIF-1α Signaling Pathway

The hypoxia-inducible factor 1 alpha (HIF-1α) is an innovative target for the action of new drugs with anti-inflammatory activity (Figure 10). Several studies with HIF-1α were performed in the last years as an attempt to explain how cells sense and to adapt to oxygen availability. These studies were recognized by the Nobel Prize of Physiology or Medicine in 2019 awarded to Kaelin, Ratcliffe, and Semenza.

Using human colon carcinoma HCT116 cells, esculetin was demonstrated to induce the hypoxia-inducible factor 1 alpha (HIF-1α), promote the secretion of vascular endothelial grown factor (VEGF), and inhibit HIF prolyl hydroxylases (PHD) activity [51]. This elegant study also suggested that catechol moiety in esculetin is required for HPH inhibition via competition with ascorbate and 2-ketoglutarate [51], given that several compounds containing catechol moieties such as quercetin and caffeic acid tend to activate HIF-1α [93,94]. In the intestinal inflammatory process, the epithelial cells provide barrier and transport functions, which are modulated by a series of physiological and morphological events such as mucus production, microvilli, and tight junctions. On the other hand, the high vascularization of intestinal tissue contributes to the counteraction of the high oxygen gradient from luminal anaerobic conditions to oxygenated tissue [95]. It has been considered that in acute and chronic inflammation, oxygen delivery, and oxygen availability or hypoxia is a key factor to trigger an inflammatory response [26,96]. The hypoxia signaling pathway is mainly coordinated by the HIF-1α stabilization, and in normoxia conditions, proline residues are hydroxylated by PHD action producing a complex with the Von Hippel-Landau (VHL) protein [97]. The complex HIF-VHL binds with ubiquitin, leading to proteasomal degradation of HIF-1α (Figure 10). Hypoxia signaling induces growth factors, such as transforming growth factor β (TGF-β) and VEGF binds with membrane-related tyrosine kinase receptors triggering a signaling pathway of phosphatidylinositol 3-kinase (PIP3K) with consequent serine/threonine-specific protein kinase 1(Akt1) phosphorylation (Figure 10) [98]. Under hypoxia, the activity of PHD is suppressed while phosphorylated Akt promotes the phosphorylation of mammalian target of rapamycin (mTOR) and FKBP-rapamycin associate protein (FRAP), regulating HIF-1α [98]. HIF-1α subunits translocate into the nucleus to bind with HIF-1β subunit and heterodimer HIF-α:HIF-β transcription factor complex then locate to the hypoxia-response elements (HRE) target genes (Figure 10), resulting in their transcriptional upregulation with the participation of coactivator p300/CREB binding protein (p300/CBP) [97].

#### 4.4.4. The Pregnane X Signaling Pathway

The pregnane X nuclear signaling pathway has been also reported as a target for the action of intestinal anti-inflammatory products, including coumarin derivatives such as imperatorin [99]. Nuclear pregnane X receptors (PXRs) are well-recognized for their function in the modulation of drug metabolism, acting as a flexible ligand for several products including drugs, natural and dietary products, hormones, and environmental pollutants [100]. Predominantly expressed in the intestine and liver, PXR after activation forms a heterodimer with the retinoid X receptor (RXR) [76]. This heterodimer binding to specific PXR response elements to control the gene expression of several proteins [76]. PXR agonists were demonstrated to attenuate intestinal inflammatory symptoms and to reduce intestinal permeability [101], improving epithelial barrier function via suppression of NF-κB expression that encoding pro-inflammatory cytokines [102]. PXR activation is a relevant antagonist of NF-κB transcriptional activity in the intestine during intestinal inflammation [103]. Imperatorin (Figure 2) mediated PXR activation suppressing the nuclear translocation of NF-κB and down-regulating pro-inflammatory production in DSS-induced intestinal inflammation in mice [99].

### 4.5. Effects of Coumarin Derivatives Intestinal Microbiota

Intestinal microbiota modulation by dietary products, mainly probiotic, prebiotic, and other natural products to improve SCFAs and other bacteria metabolites production from the fermentative process is an important approach to prevent IBD as well as to relieve symptoms of the intestinal inflammatory process. However, among all coumarin derivatives evaluated in several studies related to intestinal inflammation, only daphnetin (Figure 2) was demonstrated to act on the intestinal microbiota [74]. Daphnetin reversed DSS-induced gut dysbiosis, reducing *Bacteroides,* and increasing *Firmicutes*, which are the major SCFAs-producing bacteria [74]. Moreover, it was demonstrated that daphnetin was able to recovery zona occludens, occludin, mucin, and E-cadherin function compromised by DSS-induced intestinal inflammation, improving the intestinal epithelial integrity [74]. Using an elegant approach of the microbiota-transfer by cohousing untreated with daphnetin-treated mice, the authors reported an improvement of the clinical parameters, bacteria biodiversity, and immune response in the colon of cohousing DSS-untreated animals, when compared with DSS-inflamed mice singly housed [74]. Finally, to corroborate these data and intestinal microbiota importance in the maintenance of intestinal function, fecal microbiota from daphnetin-treated mice was transfer to mice depleted of intestinal microbiota, and the results demonstrated a remarkable improvement of disease manifestations, immune and inflammatory response when compared with the animal has received the vehicle, clearly showing that protective effects of daphnetin in intestinal inflammation, besides of its effects on the oxidative stress and immune response, were directly related of the regulation of intestinal integrity and tissue homeostasis modulated by intestinal microbiota [74]. Recently, daphnetin was also demonstrated to improve the altered intestinal microbiota composition of the glucocorticoid-induced osteoporosis rats, attenuating the intestinal barrier dysfunction [104]. Although daphnetin is the only coumarin whose intestinal anti-inflammatory activity has been directly associated with intestinal microbiota modulation, other natural and synthetic coumarin derivatives and plant extracts containing coumarins [105,106,107,108,109,110,111] were able to differentially modulate some pathogenic intestinal bacteria, but with no direct evidence and correlation with intestinal anti-inflammatory activity.

## 5. Conclusions and Perspectives

This review provided a general overview of the various coumarin derivatives with potential therapeutic applications on the intestinal inflammatory processes highlighting the ones for which the mechanism of action is at least partially defined and can serve for the design of further preclinical and clinical studies to support the use and application of coumarin derivatives as complementary therapies against IBD. In general, the mechanisms of action of coumarin derivatives observed in experimental models of intestinal inflammation and in vitro studies are similar to those described for other natural products such as flavonoids, anthocyanidins, and catechins. Although several coumarin derivatives such as paepalantine, 4-methylesculetin, daphnetin, esculetin, and osthole produce intestinal anti-inflammatory effects in lower doses when compared with other phenol compounds, it is no possible to attribute advantages in the use of these coumarins due to the lack of clinical trials and more detailed studies on efficacy and safety with these compounds. Protective effects of coumarin derivatives are related to antioxidant properties, similar to those produced by several phenolic compounds. However, some coumarins also interact with several endogenous macromolecules, different cell types, and signaling pathways as well as in innovative molecular targets. On the other hand, further studies are needed into the effects of some coumarin derivatives on the course of the disease, mechanisms of action, ability to modulated intestinal microbiota and intestinal permeability, and safety for use. Clinical trials in patients with IBD are very important to generate data for a potential application of coumarins derivatives as a complementary therapy for this chronic disease. There is scientific evidence here reported to support the suggestions of some coumarin derivatives as candidates for further pre-clinical studies and clinical trials, particularly those better studied, mechanism of action partially defined and with protective effects in lower doses, such as esculetin, 4-methylesculetin, osthole, and daphnetin.

## Figures and Tables

**Figure 1 molecules-26-00422-f001:**
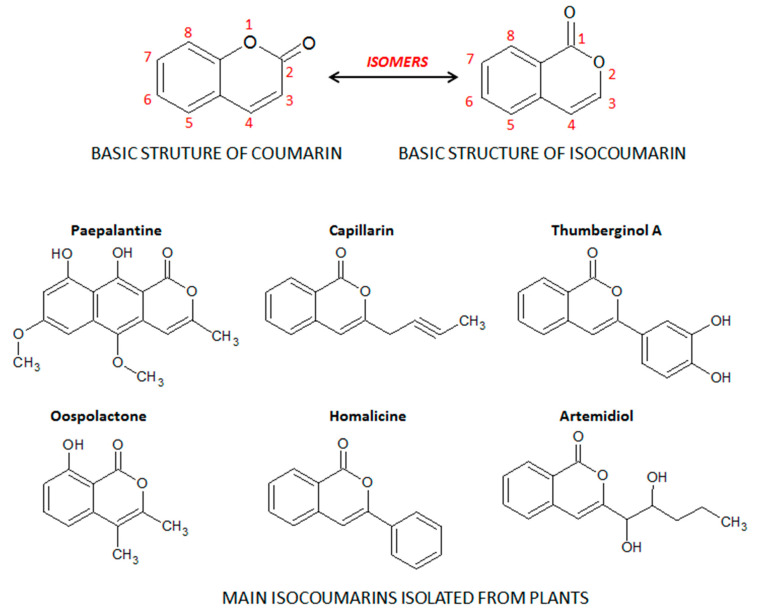
Basic structures of coumarins and isocoumarins and the main isocoumarin derivatives. Chemical structures were drawn using ACD/ChemSketch software.

**Figure 2 molecules-26-00422-f002:**
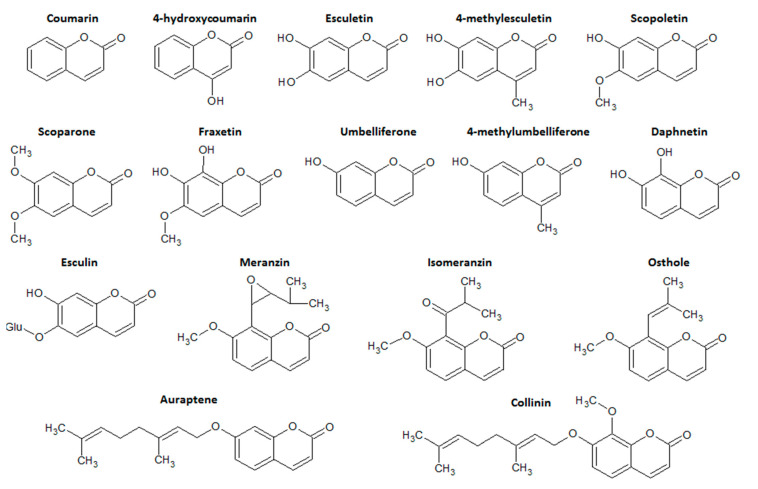
Chemical structures of the main simple coumarin derivatives with intestinal anti-inflammatory activity. Chemical structures were drawn using ACD/ChemSketch software.

**Figure 3 molecules-26-00422-f003:**
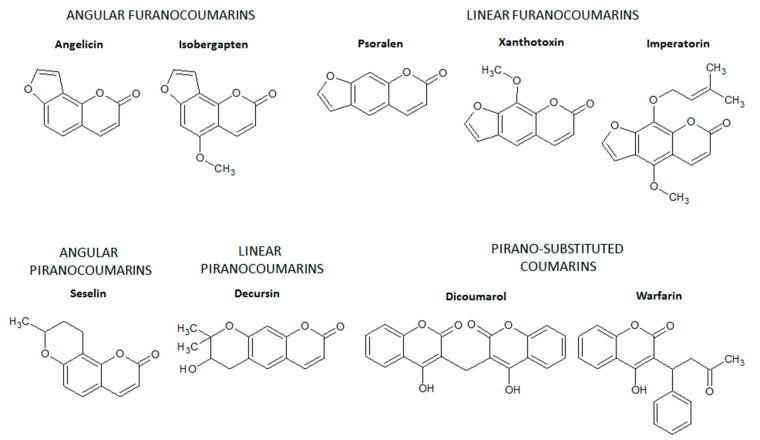
Chemical structures of angular and linear furanocoumarins, angular and linear pyranocoumarins, and pyrano-substituted coumarins. Chemical structures were drawn using ACD/ChemSketch software.

**Figure 4 molecules-26-00422-f004:**
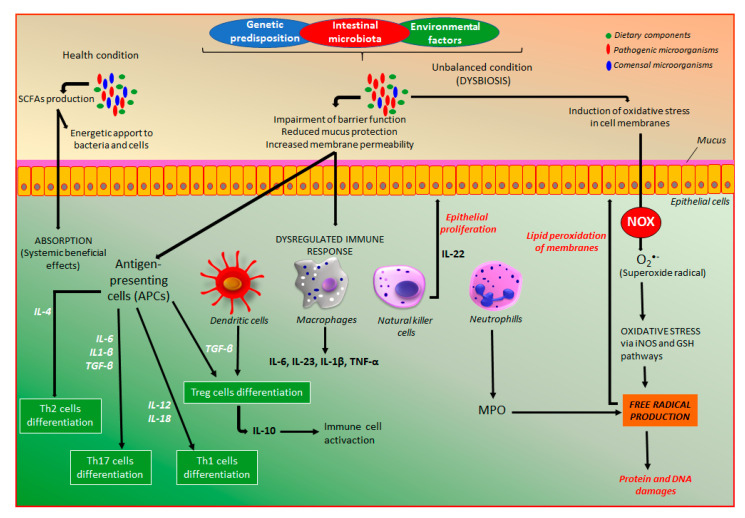
The main pathways of the intestinal inflammatory process or the action of coumarin derivatives. GSH, glutathione; IL, interleukin; iNOS, inducible nitric oxide synthase; MPO, myeloperoxidase; SCFAs, short-chain fatty acids; TGF, transforming growth factor; Th, T helper cells; Treg, T regulatory cells; TNF, tumor necrosis factor.

**Figure 5 molecules-26-00422-f005:**
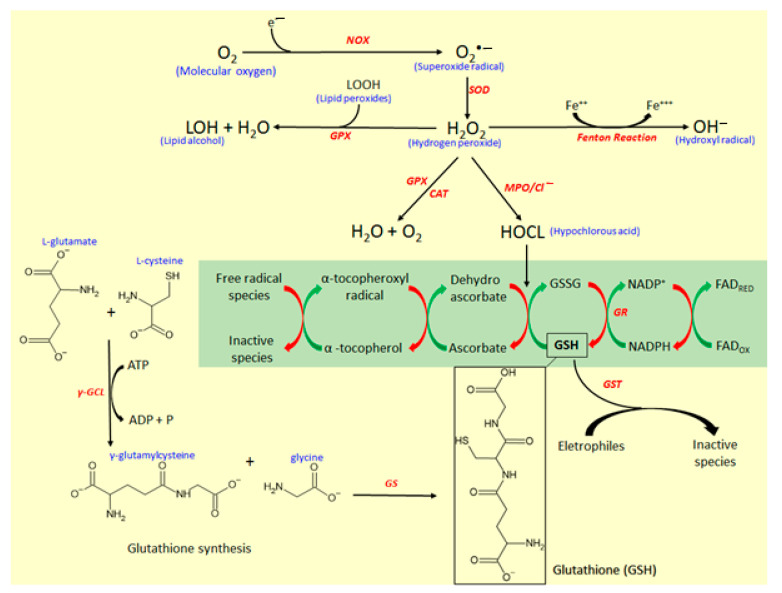
Free radical production and glutathione (GSH) antioxidant system. CAT, catalase; GPX, glutathione peroxidase, GR, glutathione reductase; GS, glutathione synthetase; GST, glutathione S-transferase; NOX, NADPH oxidase; MPO, myeloperoxidase; SOD, superoxide dismutase; γ-GLC, γ-glutamylcysteine ligase. In these oxidative processes, the following activities were reported: A. Counteraction of GSH depletion by paepalantine, coumarin, 4-hydroxycoumarin, esculetin, 4-methylesculetin, daphnetin, esculin, scopoletin, scoparone, and fraxetin; B. Inhibition of MPO activity by paepalantine, esculetin, 4-methylesculetin, daphnetin, and esculin; C. Scavenging activity of free radical by paepalantine, daphnetin, esculin, scopoletin, scoparone, and fraxetin; D. Inhibition of lipid peroxidation by esculetin, and daphnetin; E. Inhibition of GPX activity and expression by 4-methylesculetin; F. Increase of GST and GR activity and expression by 4methylesculetin. Chemical structures were drawn using ACD/ChemSketch software.

**Figure 6 molecules-26-00422-f006:**
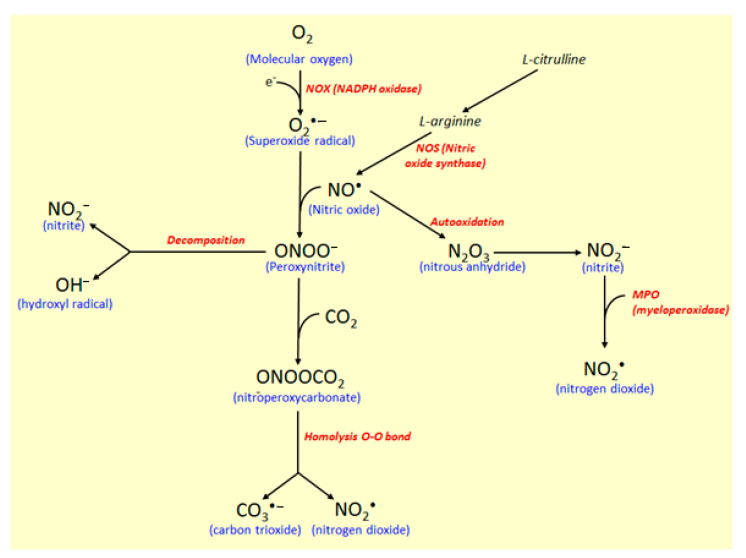
Nitric oxide synthase (NOS) pathway of free radical production. NOX, NADPH oxidase; NOS, nitric oxide synthase; MPO, myeloperoxidase. In these oxidative processes, the following activities were reported: A. Inhibition of iNOS activity by paepalantine, esculetin, esculin, auraptene, and collinin; B. Reduction of NO release by isomeranzin and esculin.

**Figure 7 molecules-26-00422-f007:**
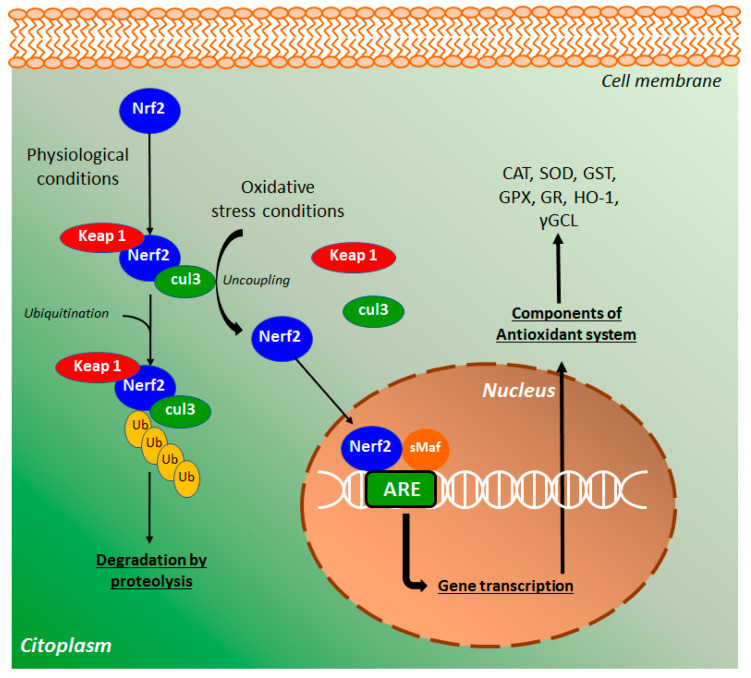
The nuclear factor erythroid 2 (NEF2)-related factor 2 (Nrf2) signaling pathway of oxidative stress. ARE, antioxidant element of response; CAT, catalase; cul3, cullin 3; GST, glutathione S-transferase; GPX, glutathione peroxidase; GR, glutathione reductase; HO-1, heme-oxygenase 1; Keap1, Kelch-like ECH-associated protein 1; Nrf2, nuclear factor erythroid 2-related factor 2; sMaf, small Maf proteins; Ub, ubiquitin; γGCL, γ-glutamylcysteine ligase. In these oxidative processes, the following activities were reported: A. Upregulation of Nrf2 by esculetin, 4-methylesculetin, daphnetin, and esculin.

**Figure 8 molecules-26-00422-f008:**
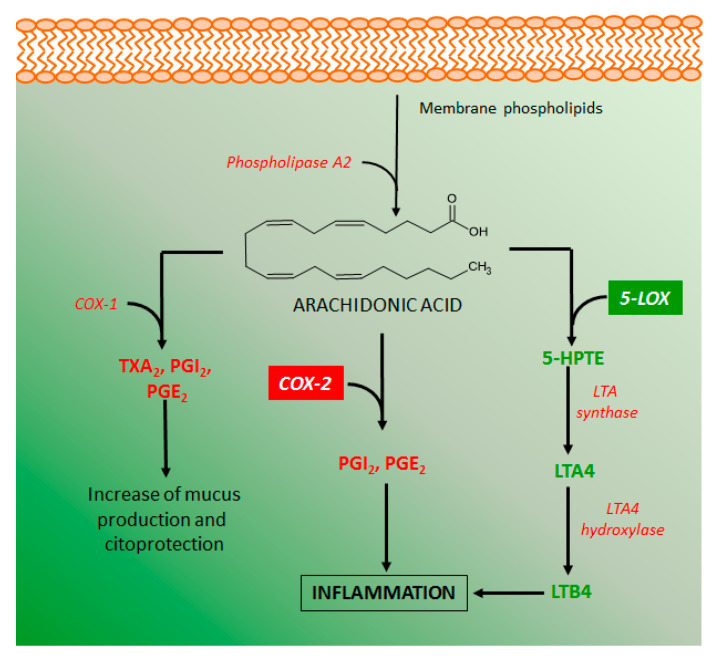
Arachidonic acid metabolism and its main pro-inflammatory mediators. 5-LOX, 5-lipooxygenase; COX-1, cyclooxygenase 1; COX-2 cyclooxygenase 2; LTA, leukotriene A; LTB, leukotriene B; PGI_2_, prostaglandin I2, PGE_2_, prostaglandin E2, TXA_2_, thromboxane A2. Inhibitory action on the arachidonic acid metabolism was demonstrated by treatment with esculetin, esculin, daphnetin, osthole, imperatorin, auraptene, collinin, and fraxetin.

**Figure 9 molecules-26-00422-f009:**
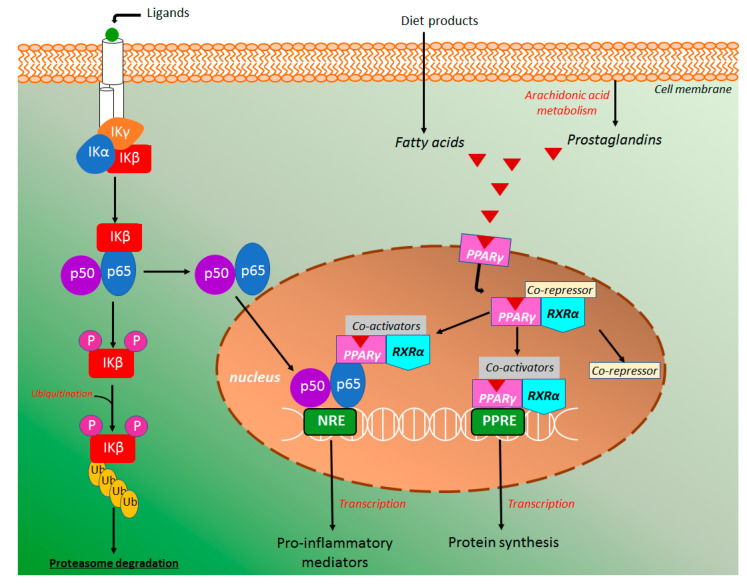
The nuclear factor-kappa b (NF-κB) and peroxisome proliferator-activated receptor gamma (PPAR-γ) signaling pathway in the intestinal inflammatory process. Modulation of NF-κB and PPAR-γ signaling pathways was demonstrated by treatment with esculetin, esculin, osthole, and isomeranzin.

**Figure 10 molecules-26-00422-f010:**
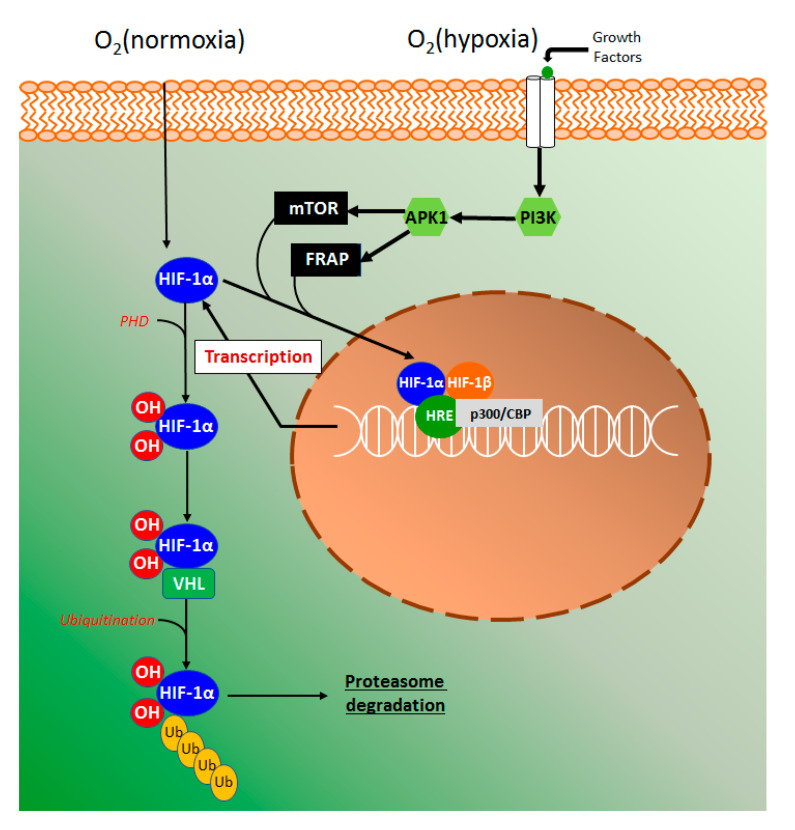
The hypoxia-inducible factor 1 alpha (HIF-1α) signaling pathway of intestinal inflammation as the target for the action of esculetin.

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
