# Peer review of "Coumarin Derivatives in Inflammatory Bowel Disease"

_molecules, 2021, doi:10.3390/molecules26020422_

Round 1

Reviewer 1 Report

This is a very interesting review about the potential role of coumarins in the treatment of intestinal inflammation, based on preclinical studies.

The review is well written, and it covers all the aspects involved in the beneficial effects exerted by these plant metabolites against intestinal inflammation. Only minor comments that can improve this review and facilitate its reading.

1. When describing the "Intestinal anti-inflammatory activity of coumarin derivatives", extensive revisions about the pathophysiological processes are included before the description of the specific effects of the differente coumarin derivatives. Maybe, these parts (lines 272-296; 311-358; 365-369; 529-542; 601-639; 665-678; and the corresponding figures) can be resumed and moved to the section "Inflammatory Bowel Diseases: general aspects". And then, when the effects of each coumarin are described, the specific mechanisms can be commented.

2. A Figure indicating the key targets for the different coumarins could be interesting to include.

3. Is it possible to establish a structure-activity relationship among the different coumarins evaluated until now?

4. Is there any advantage in the use of coumarins when comparing the known effects exerted by other phecolic derivatives, like flavonoids? A comment about this can be included.

Minor:

In lines 490 and 571 there is a typographical error: DDS instead of DSS

Author Response

REVIEWER 1

Comments and Suggestions for Authors

This is a very interesting review about the potential role of coumarins in the treatment of intestinal inflammation, based on preclinical studies.

Answer: Thank you very much for this comment.

The review is well written, and it covers all the aspects involved in the beneficial effects exerted by these plant metabolites against intestinal inflammation. Only minor comments that can improve this review and facilitate its reading.

Answer: Thank you again.

  1. When describing the "Intestinal anti-inflammatory activity of coumarin derivatives", extensive revisions about the pathophysiological processes are included before the description of the specific effects of the differente coumarin derivatives. Maybe, these parts (lines 272-296; 311-358; 365-369; 529-542; 601-639; 665-678; and the corresponding figures) can be resumed and moved to the section "Inflammatory Bowel Diseases: general aspects". And then, when the effects of each coumarin are described, the specific mechanisms can be commented.

Answer: Thank you for the comment and suggestion. I would like to inform you that I organized and changed the manuscript structure several times before its submission, including a similar version as suggested containing pathophysiology of IBD in one only section. The reviewer suggestion was the first version of the manuscript, However, I identified some problems with this organization. First, the section “IBD: general aspects” would be very long and the discussion about coumarins effects will be dislocated from the illustrative figures and IBD pathophysiology. In some cases, the text about coumarin would be very simple and displaced of the pathophysiological mechanisms. Secondly, the final manuscript organization, considering a good relation between text and figures, would be not so good. Thirdly, although there were several doubts when I was writing the manuscript relative to a final form and presentation of the manuscript, it was considered more appropriate to focus on some groups of coumarins with similar properties, for example, antioxidant coumarins, coumarins modulating the immune response, etc. The idea of the Section “IBD: general aspects” was created just for a general view of IBD, mainly definition, epidemiological data, and general aspects of IBD pathogenesis, focusing on the main target of coumarin derivatives. Besides, I identified in the comments of the second reviewer the suggestion to reduce the sections “Introduction” as well as the section “IBD: general aspects”. In the new version of the manuscript, I attempted to improve the quality of the manuscript considering the suggestions of both reviewers.

  1. A Figure indicating the key targets for the different coumarins could be interesting to include.

Answer: Thank you again. The idea of figure 4 was precisely to show a general view of the main targets for the action of coumarin derivatives. In the new version of the manuscript a general comment was included in the text and figure caption (See lines 233 to 236). Thank you very much, because this relation was not completely clear in the original version of the manuscript.

  1. Is it possible to establish a structure-activity relationship among the different coumarins evaluated until now?

Answer: Thank you very much for this comment. Although I am a pharmacologist, I have my doctoral in organic chemistry, and truly I attempted to establish some structure-activity relationship among the coumarin derivatives, which would be very interesting and elegant in this review. However, I identified that based on available data, structure-activity considerations would be very speculative because there is very variables in the published studies, particularly considering different administration routes,  doses used, different experimental models, and cell types. Although some comments about the structure-activity relationship were included in the manuscript, these considerations were based on specific studies using the same methods and similar doses and routes. As commented, it is not possible to show a detailed structure-activity consideration with available data in published articles. It would be fantastic this discussion, but using the available data,  only a hypothetical consideration and speculative would be possible.

  1. Is there any advantage in the use of coumarins when comparing the known effects exerted by other phecolic derivatives, like flavonoids? A comment about this can be included.

Answer: Thank you for the comment. In the original version of the manuscript, particularly in the Conclusion Section, a general comparison of mechanisms of action of coumarins with other phenolic compounds was reported (lines 776 to 778). Based on available data, the effects of coumarin derivatives are similar to those produced by other polyphenol compounds. Although several coumarin derivatives produce intestinal anti-inflammatory effects in lower doses when compared with other phenol compounds, it is no possible to attribute advantages in the use of coumarin because clinical trials with these compounds were not performed.

Minor:

In lines 490 and 571 there is a typographical error: DDS instead of DSS

Answer: Thank you. This was corrected in the new version of the manuscript.

Reviewer 2 Report

I red with great interest the review by Di Stasi on the potential usefulness of coumarin derivates in the treatment of IBD. The present manuscript provides a comprehensive review of the available data supporting the anti-inflammatory effect od coumarins, although the evidences mostly derives from in vitro experiments and from the animal model.

Below some specific comments:

1) I feel the manuscript will benefit from a lenght reduction in sections "1.Introduction" and "3.Inflammatory bowel diseases: general aspects" in order to promptly get to the core of the manuscript.

2) Section 3. Growing evidences support a central role of altered intestinal permeability in the onset and progression of IBD. This aspect should be stated in line 179 and highlighted throughout this section (The author may check the following references: PMID: 18829978; 30160088; 32051759) 

 3) Section 4.5. The author basically report the results from the study of Ji et al. 2019. Are there any other studies (in vitro and in vivo) investigating the effect of coumarin derivates on the modulation of microbiome and intestinal permeability?

Author Response

REVIEWER 2

Comments and Suggestions for Authors

I red with great interest the review by Di Stasi on the potential usefulness of coumarin derivates in the treatment of IBD. The present manuscript provides a comprehensive review of the available data supporting the anti-inflammatory effect od coumarins, although the evidences mostly derives from in vitro experiments and from the animal model.

Answer: Thank you very much for this comment. Yes, the scientific pieces of evidence about potential uses of coumarin derivatives in IBD are based on preclinical and in vitro studies. In the manuscript, there is a general comment about this (Conclusion Section), indicating that available data of coumarin derivatives are important precisely to stimulate further clinical trials with some of these coumarins.

Below some specific comments:

1) I feel the manuscript will benefit from a lenght reduction in sections "1.Introduction" and "3.Inflammatory bowel diseases: general aspects" in order to promptly get to the core of the manuscript.

Answer: Thank you very much for this consideration. After the complete writing of the manuscript, it was observed that the Introduction section was long. However, the idea of this section was not only to justify the review based on the importance of IBD and coumarin derivatives but also to include a general point of the view relative to non-communicable diseases, which are the most important diseases affecting the human being as well as the importance of natural products in drug discovery. In review topics with specific compounds or natural classes of compounds, generally, there is a lack of data and comments showing historical aspects of natural compounds and their effects on non-communicable diseases. Similarly, it was considered that is important the general comments of natural products research, their features, and research difficulties. Even so, an attempt to reduce the Introduction Section was performed in the new version of the manuscript. Section 3 includes introductory aspects of IBD pathophysiology to show the complex signaling pathways and endogenous mediators involved in the intestinal inflammatory process, which are the targets for the action of the coumarin derivatives. This section did not include detailed aspects of IBD pathophysiology, which were discussed in other sections of the manuscript. For example, the description of oxidative stress process was described in the section of the coumarin derivatives with antioxidant properties, the NF-κB signaling pathway was described in the section of intestinal anti-inflammatory coumarins acting on the NF-KB. The same in other sections and subsections. Besides, it was identified in the other reviewer's suggestions the inclusion of all pathophysiological aspects of IBD in this section, which would result in a very long Section 3.  Based on this, the option to maintain several aspects of the physiopathology of IBD divided into several sections, and subsections was considered a good presentation. In the new version of the manuscript, I attempted to improve the quality of the manuscript considering the suggestions of both reviewers.

2) Section 3. Growing evidences support a central role of altered intestinal permeability in the onset and progression of IBD. This aspect should be stated in line 179 and highlighted throughout this section (The author may check the following references: PMID: 18829978; 30160088; 32051759)

Answer: Thank you very much. This is a good suggestion because intestinal permeability is a key factor in the pathogenesis of IBD. In the original version of the manuscript, the dysfunctional intestinal barrier with increased permeability was reported as a key factor leading to intestinal inflammation. Based on this comment, it was included additional comments on this in the new version of the manuscript (see lines: 179, 181, 214, and 217 to 223, 783).

 3) Section 4.5. The author basically report the results from the study of Ji et al. 2019. Are there any other studies (in vitro and in vivo) investigating the effect of coumarin derivates on the modulation of microbiome and intestinal permeability?

Answer: The study of Ji et al., 2019 is the only one that evaluated a coumarin derivative in the intestinal inflammation model associating the protective effects to intestinal microbiota modulation. Several publications with natural and synthetic coumarins modulating different types of pathogenic bacteria as well as pharmacological studies using plant extracts containing coumarin derivatives were correlated to modulation of intestinal microbiota, but these data are only indirect evidence. However, it was included additional comments about this in the new version of the manuscript.